# Development of Monitoring and Mating Disruption against the Chilean Leafroller *Proeulia auraria* (Lepidoptera: Tortricidae) in Orchards

**DOI:** 10.3390/insects12070625

**Published:** 2021-07-09

**Authors:** M. Fernanda Flores, Jan Bergmann, Carolina Ballesteros, Diego Arraztio, Tomislav Curkovic

**Affiliations:** 1Instituto de Química, Pontificia Universidad Católica de Valparaíso, Avda. Universidad 330, Curauma, Valparaíso 2340000, Chile or fflores@agroadvance.cl (M.F.F.); jan.bergmann@pucv.cl (J.B.); 2Agroadvance SpA. Camino Melipilla, Peñaflor, Santiago 9710000, Chile; 3Facultad de Agronomía e Ingeniería Forestal, Pontificia Universidad Católica de Chile, Avda. Vicuña Mackenna 4860, Macul, Santiago 7820436, Chile; caballesteros@uc.cl; 4Facultad de Ciencias Agronómicas, Universidad de Chile, Avda. Santa Rosa 11315, La Pintana, Santiago 8820808, Chile; dsarrazt@uchile.cl

**Keywords:** pheromone, field trials, moth phenology, vineyards, apple orchards, blueberry orchards, SPLAT wax matrix, remaining pheromone in point sources

## Abstract

**Simple Summary:**

*Proeulia auraria* is a native and growing pest insect in fruit orchards in Chile, which calls for environmentally friendly management methods. Using synthetic pheromone compounds, we conducted field trials to optimize the septa load for monitoring adult moths. Using the optimized blend we studied the phenology of males in vineyards, apples, and blueberries, finding two large flight cycles lasting from September to May. Afterward, based on field trials, we concluded that 250 point sources loaded with a total of 78 g/ha of the pheromone blend, provided high disruption of male-female encounters for mating in all crops tested for at least 5 months. We concluded that mating disruption is feasible for *P. auraria*, needing now the development of a commercial product and of protocols to control this pest.

**Abstract:**

The leafroller *Proeulia auraria* (Clarke) (Lepidoptera: Tortricidae) is a native, polyphagous, and growing pest of several fruit crops in Chile; it also has quarantine importance to several markets, thus tools for management are needed. Using synthetic pheromone compounds, we conducted field trials to optimize the blend for monitoring, and to determine the activity period of rubber septa aged under field conditions. We concluded that septa loaded with 200 μg of E11-14:OAc + 60 μg E11-14:OH allowed for efficient trap captures for up to 10 weeks. Using this blend, we studied the phenology of adult males in vineyards, apple, and blueberry orchards, identifying two long flight cycles per season, lasting from September to May and suggesting 2–3 generations during the season. No or low adult activity was observed during January and between late May and late August. Furthermore, mating disruption (MD) field trials showed that application of 250 pheromone point sources using the dispenser wax matrix SPLAT (Specialized Pheromone and Lure Application Technology, 10.5% pheromone) with a total of 78 g/ha of the blend described above resulted in trap shutdown immediately after application, and mating disruption >99% in all orchards for at least 5 months. We concluded that MD is feasible for *P. auraria*, needing now the development of a commercial product and the strategy (and protocols) necessary to control this pest in conventional and organic orchards in Chile. As far as we know, this is the first report on MD development against a South American tortricid pest.

## 1. Introduction

The Chilean fruit or pear leafroller, *Proeulia auraria* (Clarke) (Lepidoptera: Tortricidae: Tortricinae), is distributed in Chile, from Atacama (coordinates −27.37, −70.33) to Los Lagos (−41.47, −72.94) [1,2,3]. The species is also mentioned in the literature from Argentina but restricted to Chile based on the Centre for Agricultural and Bioscience International [4,5], therefore considered a native moth. *Proeulia auraria* is the economically most important species of the Genus *Proeulia,* endemic to Chile [6,7,8]. It is a polyphagous insect that has moved from native hosts to exotic plants [9]; the hosts reported for either *P. auraria* or *Proeulia* sp, include citrus, figs, grapes, kiwifruits, loquats, pome fruits, pomegranates, stone fruits, and walnuts; several native plants (e.g., *Ugni molinae* Turcz), weeds (e.g., *Galega officinalis* L.), and ornamental trees (e.g., *Platanus orientalis* L.) [3,9,10,11,12]. During the last four decades, *P. auraria* has become a key pest in some vineyards and orchards in Chile [3,10]. This is probably due to changes in management tactics for other pests (e.g., withdrawal of insecticide applications because of increasing use of mating disruption against *Cydia pomonella* L. and *Lobesia botrana* (Denis & Schiffermüller)), the decreasing amount of natural enemies due to insecticide use, and/or replacement of the original vegetation including *P. auraria* native hosts [9,10]. The larvae cause damage by folding foliage and eating flowers, reducing the photosynthetic potential, the canopy development, and the fruit set. Larvae also feed on the fruit epidermis, and cause surface tunneling, which makes the fruits unmarketable, and facilitates the development of saprophagous insects (e.g., *Drosophila melanogaster* Meigen) or fungal diseases (e.g., *Botrytis cinerea* Pers.) [9]. However, the most important economic impact is due to its quarantine status, frequently causing rejections of Chilean fresh fruits for export when immature stages (mainly larvae) are found during fruit inspections after harvest [10,11]. In fact, several reports [13,14] mentioned *P. auraria* within the list of new pests representing a risk of introduction to different new areas importing plants and fresh fruits from South America. For this reason, many growers currently rely on different insecticides (e.g., *Bacillus thuringiensis*, organophosphates, etc.) as the most common pre-harvest tactic used against this pest [15,16].

Monitoring *P. auraria* males for timing insecticide applications has been conducted in Chile using pheromone-baited traps [16]. The first report of *P. auraria* pheromone established a 7:3-ratio of (*E*)-11-tetradecenyl acetate (E11-14:OAc) and (*E*)-11-tetradecenol (E11-14:OH) as the main components [17], whereas Roelofs and Brown [18] reported only the alcohol as the sex pheromone. During several years, *P. auraria* monitoring in Chile was conducted using pheromone lures for the tufted apple bud moth (TBM), *Platynota ideausalis* Walker (Tortricidae) [10] containing a 1:1 ratio of E11-14:OAc and E11-14:OH. However, a significantly more attractive 4-component sex pheromone for *P. auraria* populations from central Chile was reported by our group [19]: tetradecyl acetate (14:OAc), E11-14:OAc, (*Z*)-11-tetradecenyl acetate (Z11-14:OAc), and E11-14:OH, in a relative ratio of 11:100:1:37. In fact, recently this blend was successfully tested in Southern areas of Chile [20]. Additionally, preliminary data obtained from small-scale field trials suggested that *P. auraria* males might be susceptible to mating disruption [21].

Based on that information, the objectives of this study were to develop and optimize monitoring and to further explore the potential of mating disruption (MD) for *P. auraria*. Particularly, we conducted field trials to determine the optimum blend and the longevity of septa for efficient monitoring, and the initial load and emitter density necessary to maximize disruption.

## 2. Materials and Methods

### 2.1. Orchards

Field trials were conducted in commercial fields in the Region of O’Higgins, central Chile: a conventional 5-years-old blueberry orchard (cvs. O’Neill and Brigitta planted at 3 × 0.8 m) near San Francisco de Mostazal (coordinates −33.98, −70.68); two organic (>20 years old) vineyards (cvs. Cabernet Sauvignon, Carmenere, and Syrah at 2–2.5 × 1–1.3 m) near Requinoa (−34.28, −70.81) and Nancagua (−34.63, −71.22); and an organic 10-years-old apple orchard (cvs. Granny Smith and Gala at 2.7 × 2 m) near San Fernando (−34.59, −70.98). All four orchards have historically had infestations by *P. auraria*, thus, spray programs using *Bacillus thuringiensis* (Bt) and/or Spinosad were being applied in all of them. However, neither Bt nor Spinosad were applied in the MD plots during our study.

### 2.2. Chemicals, Traps, and Pheromone Matrix

E11-14:OAc (purity 99%, containing ca. 1% of Z11-14:OAc), Z11-14:OAc (99%), and E11-14:OH (>99%) were purchased from Bedoukian Research Inc (Danbury, CT, USA) and were either used as received (hereafter “NP” or “non-purified”) or were purified by column chromatography on silica gel impregnated with silver nitrate to remove the respective geometric isomer and other impurities (hereafter “PU” or “purified”). The isomeric purity of the compounds after purification was >99%, as determined by gas chromatography. Tetradecyl acetate (14:OAc) was obtained according to the procedure described previously [19] and had a purity of >99%. All compounds were dissolved in hexane (Suprasolv^®^, Merck, Darmstadt, Germany), and appropriate amounts were loaded on white rubber septa (Sigma Aldrich) to prepare the treatments described below. Septa for control treatments were loaded with hexane alone. After evaporation of the solvent, septa were placed inside Pherocon VI Delta traps (Trécé Inc., Adair, OK, USA) which were hung at canopy level (1.5–1.8 m) separated 30 m from each other for composition, proportion, and septa aging trials, or separated >50 m (for phenology and mating disruption studies). Unlike the vineyard and the blueberry orchard, the plots used in the MD trial in the apple orchard were uprooted just after harvest (late February) and the respective traps destroyed, preventing us to complete that study. Traps were placed at the south-west quadrant inside the tree canopy, considering always a distance of ca. 10 m from the plot edges. For mating disruption trials, an emulsified microcrystalline wax matrix called SPLAT^®^ (an acronym for “Specialized Pheromone and Lure Application Technology”, [22]) was mixed with *P. auraria* pheromone components (10.5% *w*/*w* of a blend consisting of non-purified E11-14:OAc and E11-14:OH at a 1:0.3 ratio), loaded in manual silicone sealant guns, and stored at −20 °C until application in the fields. The mixture was prepared by ISCA Technologies (Riverside, CA, USA). SPLAT not loaded with pheromone was used as a control treatment in the mating disruption trials, as also recently reported by other authors [23,24].

### 2.3. Optimizing Pheromone Load in Septa

The first experiment studied the composition of lures. Treatments for this field trial (“composition trial”) were: (1) E11-14:OAc (100 μg), E11-14:OH (30 μg)—both NP -, and 14:OAc (14 μg); (2) E11-14:OAc (100 μg) and E11-14:OH (30 μg)—both NP -; (3) E11-14:OAc (100 μg) and E11-14:OH (30 μg)—both PU -; (4) E11-14:OAc (100 μg), E11-14:OH (30 μg), Z11-14:OAc (1 μg)—all PU -, and 14:OAc (14 μg); (5) control (only hexane). This experiment was carried out in the Requinoa vineyard starting 12 November 2014. The second experiment evaluated the effect of the two main pheromonal compounds ratio. Treatments for this field trial (“proportion trial”) were blends of E11-14:OAc and E11-14:OH (both NP) at ratios (μg/septum): (1) 100:100; (2) 100:60; (3) 100:30; (4) 100:10; (5) hexane only. This experiment was carried out in the Nancagua vineyard starting 18 February 2015. In both trials, the traps were deployed in a randomized complete block design with four replicates. Blocks, and traps within blocks, were at least 30 m apart from each other. Traps were checked for captures 7 days after setup in the field. Results are expressed as the average cumulative catches/trap (*n* = 4) ± SE. Data were analyzed by straight 2-way fixed-effect ANOVA and the Tukey HSD test for mean discrimination (*p* ≤ 0.05) [25]. Statistical analyses were done with MINITAB version 2019.

To evaluate lures longevity, rubber septa (*n* = 30/treatment) were loaded with mixtures of E11-14:OAc and E11-14:OH (both NP) as follows: (1) 50 µg + 15 µg, (2) 200 µg + 60 µg, (3) 800 µg + 240 µg, (4) hexane only (control). Afterward, twelve septa (3 per treatment) were placed in a 6 × 4.5 cm tulle bag (*n* = 10). Thus, on 17 December 2014, all bags were placed on vines in the vineyard (Nancagua) at ca. 1.5 m height and ca. 10 m apart, and were covered by a piece of corrugated cardboard (as a roof) to avoid direct sun exposure. During up to 10 weeks, one bag per week was removed and maintained in a sealed container at −20 °C until use. On 18 February 2015, septa were placed inside Delta traps (1 septum per trap) which were deployed in the field in a completely randomized block design (*n* = 3), and male catches were counted 5 days later. Trap distribution and setup were as described previously. Data were analyzed by General Linear Model-ANOVA with a factorial arrangement and means were separated by the Tukey HSD test (*p* ≤ 0.05) [25].

### 2.4. Remaining Pheromone in Septa after Aging under Field Conditions

To determine the amount of remaining pheromone after aging septa under field conditions, thirty rubber septa were loaded with 200 µg of E11-14:OAc and 60 µg of E11-14:OH (both non-purified), and 5 septa were placed together in tulle bags (as described above) in the Nancagua vineyard on 12 October 2016. One bag was immediately taken to the freezer and the remaining 5 bags were kept under field conditions as described above. Every 30 days another bag was collected and stored in a sealed container at −20 °C until 10 March 2017. The remaining pheromone was extracted from individual septa with 5 mL hexane for 30 min assisted by ultrasound. Extracts were analyzed by gas chromatography-mass spectrometry (GC-MS), using a Shimadzu GCMS-QP2010 Ultra equipped with a fused silica RTX-5 capillary column (30 m × 0.25 mm id, 0.25 μm film, Restek) and a temperature oven program from 50 °C for 5 min to 270 °C at 8 °C min^−1^. Quantification was based on the area under the respective peak, compared to an external calibration curve obtained using authentic standards.

### 2.5. Proeulia Auraria Adult Male Phenology

Monitoring of *P. auraria* males was conducted during two consecutive seasons (2015/16 and 2016/17) in one vineyard (Requinoa) and the two orchards described above. Septa were loaded with 200 µg of E11-14:OAc and 60 µg of E11-14:OH (both NP) and replaced every 10 weeks. Delta traps (*n* = 3/orchard or vineyard) with those septa were installed in 2015: monitoring started at the beginning of September (vineyards and apples) or October (blueberries) and ended in May 2017. The numbers of trapped males were counted once a week, except between mid-June and late August, when traps were revised once or twice a month.

### 2.6. Proeulia Auraria Mating Disruption Trials

The SPLAT was applied in 30 mL plastic cups (to prevent the material from running off) in amounts equivalent to 78 g/ha (MD treatment; SPLAT 10.5% *w*/*w* of E11-14:OAc and E11-14:OH both NP, at 1:0.3 ratio) and 0 g/ha (control treatment; SPLAT pheromone-free), with 250 point sources/ha (based on previous results from small scale field trials [21]). At the beginning of September 2016, cups were homogeneously distributed in the plant canopy (1.5–1.8 m height) in the respective plots (ca. 4 ha each); plots were spaced at least 50 m apart in the Requinoa vineyard and the orchards. Two pheromone-baited Delta traps (as described above, see *Proeulia auraria adult male phenology*) were placed in July 2016 in each plot, and captured males were counted weekly until March 2017. Disruption indexes (DI) for seasonal male captures/treatment were calculated for each fruit species, using the equation DI = 100 * (C − D)/C (where C = captures in control plots, D = captures in respective MD plots).

Fruit damage was evaluated at harvest by contrasting two randomly chosen fruit clusters per tree, in 150 (blueberries), 421 (apples), and 559 (grapes) plants/plot, in the control and the MD treatment, respectively. The results were analyzed using the test for the equality of proportions (*p* ≤ 0.05) [25].

### 2.7. Remaining Pheromone in SPLAT Matrix after Ageing under Field Conditions

To determine the amount of remaining pheromone in the SPLAT matrix after aging under field conditions, 30 plastic cups containing the pheromone-loaded matrix as described above were placed in the Requinoa vineyard on 15 October 2016. After 30, 60, 90, 120, 150, and 210 days, 5 cups were removed at a time and submitted to analysis. The remaining pheromone was extracted as described above. After filtration through grade 1 filter paper, extracts were analyzed and quantified as described above (see *Remaining pheromone in septa after aging under field conditions*).

## 3. Results

### 3.1. Optimizing Pheromone Load in Septa

#### 3.1.1. Pheromone Composition Trial

The field trial designed to evaluate optimum pheromone composition showed significantly larger *P. auraria* male catches (*F =* 24.21, *df* = 4, *p <* 0.001, Figure 1) when either the 2- or the 3-component blends, prepared with NP compounds, were used. Treatments using blends prepared with purified compounds were statistically less attractive. All blends were significantly more attractive than the control.

#### 3.1.2. Pheromone Proportion Trial

Results from a field trial designed to identify the most attractive ratio of E11-14:OAc and E11-14:OH, showed that the largest *P. auraria* captures (*F =* 215.78, *df* = 3, *p* < 0.001, Figure 2) were obtained with 100 μg of E11-14:OAc and either 10 or 30 μg of E11-14:OH (1:0.1 or 1:0.3 ratios). Treatments with lower ratios (1:0.6 or 1:1) provided significantly lower captures. No catches occurred with the control.

#### 3.1.3. Septa Longevity

The field trial designed to evaluate the longevity of septa loaded with three different amounts of E11-14:OAc and E11-14:OH (1:0.3 ratio), aged for up to 10 weeks, showed significant interaction (pheromone load x aging time; *F* = 16.57, *df* = 18, *p <* 0.001), so both factors did not act independently. Thus, we perform the contrast with the Tukey test for every week x pheromone load (Table 1, SE = 1.28). Results shown in most cases (7 out of 10) 50 μg of E11-14:OAc + 15 μg E11-14:OH significantly lower captures, despite some week-to-week variability. On the other hand, 200 μg of E11-14:OAc + 60 μg E11-14:OH was equal to 800 μg of E11-14:OAc + 240 μg E11-14:OH also in seven cases, in two cases the former treatment was greater, and only in one the latter treatment was greater. Based on that, we selected the 200 μg of E11-14:OAc + 60 μg E11-14:OH pheromone load for further studies, considering a septa replacement time of 10 weeks. No males were captured in control traps.

### 3.2. Remaining Pheromone in Septa after Ageing under Field Conditions

The amount of pheromone remaining in septa originally loaded with 200 µg E11-14:OAc + 60 µg E11-14:OH, and aged up to 5 months under field conditions, decreased over time, reaching close to 38% of the initial amount (the acetate and the alcohol) after 150 days, with a release rate equivalent to 12%/month. Analysis of the individual components showed that the initial proportion of 1:0.3 (at day 0) had increased to 1:0.1 five months later (Figure 3).

### 3.3. Proeulia Auraria Adult Male Phenology

Regarding *P. auraria* phenology, adult male catches in the vineyard and the orchards showed similar patterns in both seasons (2015–2016 and 2016–2017), with two very long flight cycles, the first one from late September until early January, and the second one from early February through early/mid-May, in three localities from the O’Higgins region (Figure 4). Very low or no male adult catches occurred during January, and from early/mid-May until late August or early September.

### 3.4. Proeulia auraria Mating Disruption Trials

During the MD field trials conducted in all orchards, *P. auraria* catches and disruption showed a similar trend (Figure 5A–C). Captures started at least a couple of weeks before the application of SPLAT (mid-September) in both plots (MD and control), except in apples, where no catches in the MD plot were observed during that period, probably due to a more delayed rise in male activity in the area (as shown in Figure 5B). One week after the application of pheromone-loaded SPLAT, the captures fell to zero (trap shutdown) in all disruption plots, whereas captures continued in the control plots. During the following months, male captures continued to be almost nonexistent in MD plots (0 in blueberries and apples, and only 8 in grapes between September–March/April), whereas relatively high numbers of males were caught in the respective control plots during the season (apples: 292, blueberries: 343; and vines: 838 males). Thus, disruption over the season reached 100% in blueberries and apples, and 99.1% in grapes. The marginal reduction of disruption in grapes occurs essentially in the last week of April.

At harvest, no fruit damage was observed in blueberries. In the apple orchard, the percentage of damaged fruit was not significantly different between the MD plot (3.39%) and the control plot (3.95%) (*p* = 0.6). In grapes, the MD plot had 5.99% of damaged fruits, while in the control plot, 3.48% of fruits were damaged. These values are significantly different (*p* = 0.005).

### 3.5. Remaining Pheromone in SPLAT Matrix after Aging under Field Conditions

The amount of pheromone remaining in SPLAT originally loaded with 3 g/cup of 10.5% *w*/*w* E11-14:OAc and E11-14:OH at 1:0.3 ratio, and aged up to 7 months under field conditions, decreased over time for both components, reaching close to 29% of the initial loading after 150 days and keeping close to the original ratio during that period of time. On the other hand, the evaluation at 210 days showed 82% reduction from the original load and a significant deviation from the original ratio (Figure 6).

## 4. Discussion

The results from our studies designed to optimize the pheromone composition load in septa for *P. auraria* male attraction to traps are generally consistent with our previous reports about optimum composition, where the 4-component blend (in a 100:37:4:11 ratio) attracted more males than several incomplete combinations [19]. Furthermore, the synergistic effect of small amounts of the geometric isomer (Z11-14:OAc) of the main compound (E11-14:OAc) was also confirmed in our present experiments. This is of particular interest for applications in the field, because commercially available E11-14:OAc usually contains small amounts of the geometric isomer. As long as this small impurity does not exceed 1%, a time-consuming purification process is not only not necessary, but even would be detrimental, as the synergist would be removed. It has been discussed in the literature whether or not the whole blend is needed for moth monitoring or mating disruption, or if a partial blend is sufficient [26]. In fact, most monitoring or control strategies using pheromones are conducted only with the main pheromone component, despite the fact that many more chemicals have been identified in most tortricid pheromone blends [27], but there are other examples [28,29,30] for successful MD studies using binary blends against lepidopteran pest species. On the other hand, there are also some reports on the effects of impurities (including those produced by isomerization from the main component within the septum) modifying attractiveness as well [31]. In the case of *P. auraria*, we selected an incomplete 2-component blend (E11-14:OAc and E11-14:OH), considering its efficiency, ease of use (no further purification needed), and lower cost.

The results from our trial comparing different proportions of E11-14:OAc and E11-14:OH are not fully comparable to previous studies since the 1:0.1 and the 1:0.6 ratios were not tested before. On the other hand, the 1:1 ratio was previously tested [19], but this blend was never the most attractive treatment in field trials, despite the fact it is the ratio used in a lure commercialized in Chile for *P. auraria* monitoring [10]. Our results mostly agree with previous work where a ratio of 1:0.37 [19] was usually the most attractive treatment among several multi-component blends tested. Thus, considering its efficiency and previous data, the 1:0.3 ratio was selected for the next studies. Regarding the results obtained in our study to determine optimum septa replacement timing, we concluded septa loaded with 200 μg of E11-14:OAc + 60 μg E11-14:OH replaced every 10 weeks was more convenient for monitoring *P. auraria*. The duration of lures targeting tortricids for monitoring is very variable (from a few weeks to several months), depending mostly on the dispenser type and pheromone loads [32]. Some studies have concluded, similar to our results [33], a 10-week period for septa replacement when loaded with similar compounds for trapping tortricids in vineyards. The remaining pheromone in septa over time showed a behavior approaching the first kinetic dynamic described for this type of chemicals evaporating from similar dispensers [34], which would explain the relatively long activity observed in our trials. Considering the aforementioned results, a blend of 200 µg of E11-14:OAc and 60 µg E11-14:OH (1:0.3 ratio) was selected, with replacement of the septa every 10 weeks, for *P. auraria* monitoring.

The monitoring of the flight activity of *Proeulia auraria* based on male captures in pheromone-baited traps suggests at least two generations during the season in central Chile. The second-generation complete development the following season, since winter populations (June–August) are composed mostly of second, third, and fourth larval instars. These larvae overwinter (with low activity) in refuges [10], and the occurrence of overlapping stages [9] led to a large period of adult emergence and flight during next spring, as was observed in our results. This pattern differs from other tortricid species affecting the same crops in Chile, for example, *C. pomonella* and *L. botrana*, which spend the winter as a uniform population consisting of a single instar/stage [35,36]. Thus, in the case of *P. auraria*, it is necessary to consider a complementary treatment against immatures along (or before) the installation of mating disruption targeting adult males.

Our results show that male captures in traps tend to be greater in the vineyard, medium in the apples, and lower in the blueberry orchard, being lesser in the second consecutive study season in all localities. Previous studies [1,2] conducted in the Metropolitan region (100–150 km to the north from our tested area), mostly in pear orchards using the TBM lure (the tufted apple bud moth, *P. ideausalis*), showed similar results to those reported here, but with shorter flights and events occurring a couple of weeks earlier. Those reports also suggested the occurrence of two generations per season for *P. auraria*. It should be mentioned that up to three generations for *P. auraria* have been estimated for perennial orchards such as oranges, or in seasons with warmer autumns in central Chile [1,10]. In general, the first generation of adults and its offspring are present in orchards during sprouting/bloom, whereas the second generation originates most immatures which are present between fruit set and harvest, and these juveniles (mainly larvae) are the reason for detections occurring during post-harvest inspections for fruit species.

Previous preliminary studies on mating disruption (MD) carried out in small plots per treatment (0.1 ha/plot, [21]) suggested that *P. auraria* was disruptible with the pheromone deployed in the SPLAT matrix. Those results oriented us to define the pheromone load per point source (78 g pheromone/ha) and the source density (250/ha) reported herein. In the present study, we conducted trials at a larger scale (4 ha/plot), aiming at the validation of these parameters. The values of both parameters -pheromone dose and SPLAT point source density, per ha- are within the ranges reported for other tortricids using the same matrix for MD purposes [37,38]. The trap shutdown in all MD plots in our trials was also observed in other studies conducted to develop mating disruption against different tortricids [39,40]. The very high levels of disruption obtained in our experiments (99–100%) suggest high efficiency and are above the values reported (77–89%) in several similar reports [41,42,43,44]. The marginal reduction of disruption in grapes occurred mostly in the last week of April, almost seven months after pheromone application, which is longer than the usual protection required for MD formulations (e.g., 180 days for *L. botrana*) used against tortricids [36]. Considering the varieties used in our trials and the respective localities, where the latest harvest time is usually in March, the MD tested formulation would cover the whole season. This reduction in disruption in grapes occurred when the remaining amount of pheromone in the SPLAT matrix was reduced by 82% and the original components ratio was significantly changed from 1:0.3 to 1:0.5, which we demonstrated previously to be less efficient to maximize attraction [19]. Thus, it seems to be important to develop a dispenser matrix able to keep the proportion of compounds as close to the original as long as possible (i.e., covering the whole period of male flights) for efficient disruption. The possible mechanism for mating disruption in *P. auraria* has not been identified. The literature indicates that the mechanisms in tortricids are either non-competitive (e.g., sensory adaptation) or competitive based on dispenser attraction (e.g., false-trail following) [44], being the latter cited for many tortricid species, including the target of that study (*Cydia* (*Grapholita*) *molesta* (Busck)) [22]. However, researchers in the same group, in a subsequent study [45], concluded that a competitive mechanism acted on *C. molesta* when using monitoring lures (with a low pheromone load per septum) for mating disruption, but a non-competitive one operated when mating disruption dispensers (with the equivalent to 1500× the release rate from monitoring lures) were used, demonstrating a switch of the disruption mechanism depending on the atmospheric pheromone concentration.

Regarding the level of fruit damage at harvest, in apples, no differences occurred between the control plot (sprayed with Bt and/or Spinosad) and the MD plot (no other control methods applied), whereas in grapes the damage was significantly greater in the MD plot. Similar results have been reported previously, where high levels of mating disruption on males (as observed in our field trials) do not necessarily match with the absence of damage in crops because fertilized females can migrate from nearby areas [46], particularly if the MD plot is within an orchard with high pest pressure (as we observed at the Requinoa vineyard). Thus, mating disruption treatments should consider covering the whole orchard in order to properly evaluate crop damage when using this technique.

## 5. Conclusions

In conclusion, our results set the basis for the development of more sustainable management of this important pest in Chile. We established an optimized blend of pheromone compounds for monitoring based on commercially available material and also established that the pheromone lures (septa) remain attractive for at least 10 weeks. The information generated about male flight dynamics will help to develop a phenological model for *P. auraria* for better timing of insecticide application or installation of mating disruption dispensers. As far as we know, our findings on mating disruption against *P. auraria* are the first report for a South American tortricid affecting vineyards/orchards and demonstrate that it is possible to confuse males of this increasing pest using this technique. However, there is still the need to produce a commercial product and to develop protocols for the application in vineyards and orchards in Chile. The challenge of future work is: to test the technology on larger scales (whole vineyards/orchards) and to evaluate the respective fruit damage level; to determine the effects on other synchronic and sympatric pests and natural enemies; to evaluate the economic feasibility for a commercial product development [47] given the relatively small size of the Chilean market, and to adapt the technology to the particular cycle of *P. auraria*. However, once developed, this will be a valuable tool for local, particularly organic growers not having many efficient options to manage this pest or those who look for new and more environmentally friendly tactics against *P. auraria*.

## Figures and Tables

**Figure 1 insects-12-00625-f001:**
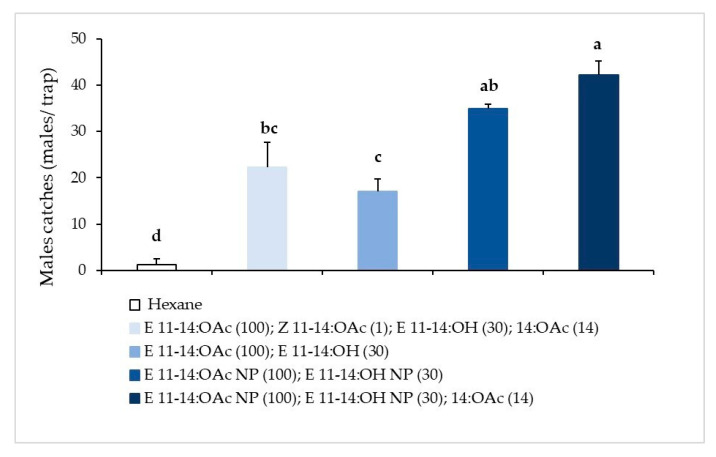
Male *P. auraria* cumulative catches (avg. ± SE) in traps (*n* = 4) loaded with different pheromone compositions or hexane (control), Requinoa, central Chile, 12–19 November 2014. Different small letters above a column mean significant differences between treatments. NP = commercial compounds not purified; if not indicated, the compounds were purified. Figures in parenthesis represent μg/septa.

**Figure 2 insects-12-00625-f002:**
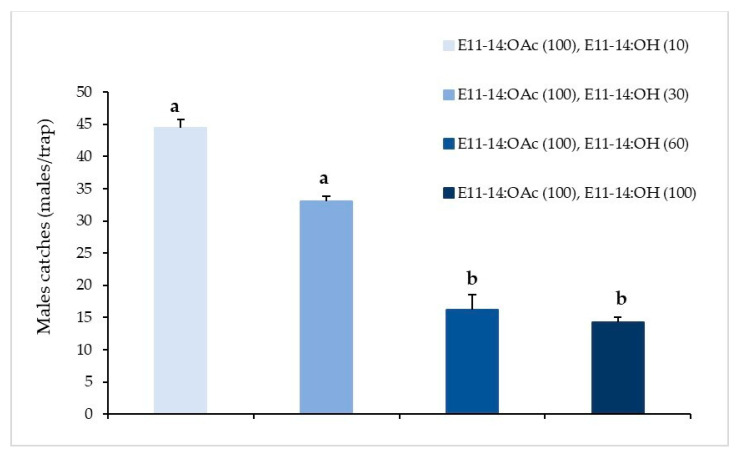
Male *P. auraria* cumulative catches (avg. ± SE) in traps (*n* = 4) loaded with different ratios of E11-14:OAc and E11-14:OH. Nancagua, central Chile, 18–25 February 2015. Different small letters above a column mean significant differences between treatments. Figures in parenthesis represent μg/septa.

**Figure 3 insects-12-00625-f003:**
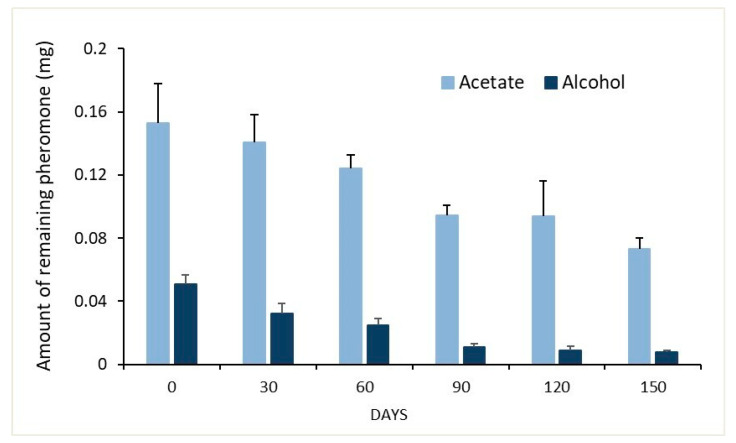
Amount of pheromone remaining (µg ± SE) in septa (original load of 200 µg E11-14:OAc and 60 µg E11-14:OH) aged up to 150 days under field conditions from October 2016 through March 2017.

**Figure 4 insects-12-00625-f004:**
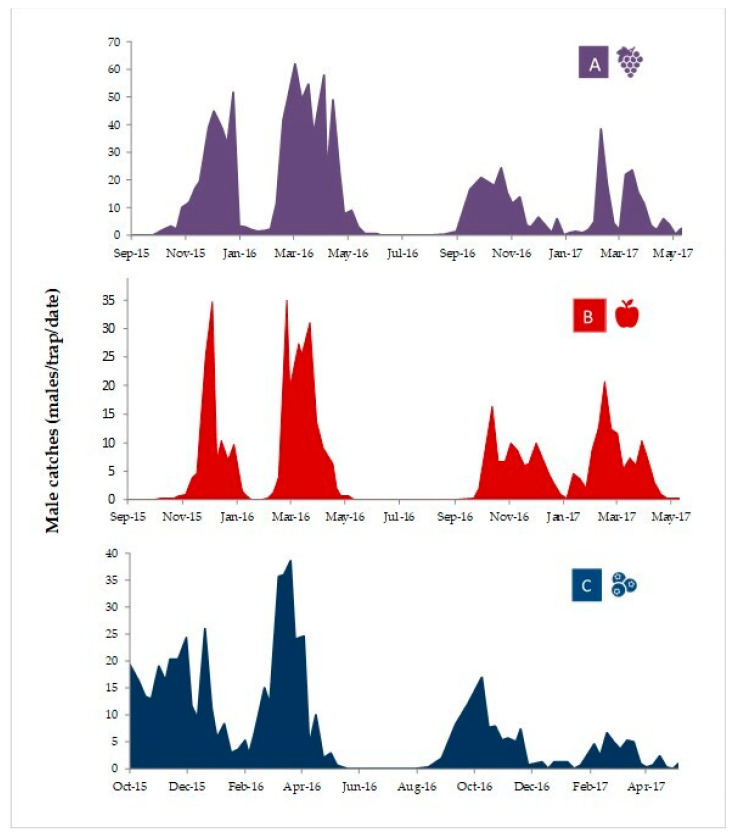
Captures of *P. auraria* (total males/3 delta traps/date) in a vineyard (**A**: Requinoa), an apple orchard (**B**: San Fernando), and a blueberry orchard (**C**: San Francisco de Mostazal), all located in the O´Higgins region, central Chile, during two consecutive seasons (September (**A**,**B**)/October (**C**) 2015 through May 2017), using septa loaded with 200 µg of E11-14:OAc and 60 µg of E11-14:OH.

**Figure 5 insects-12-00625-f005:**
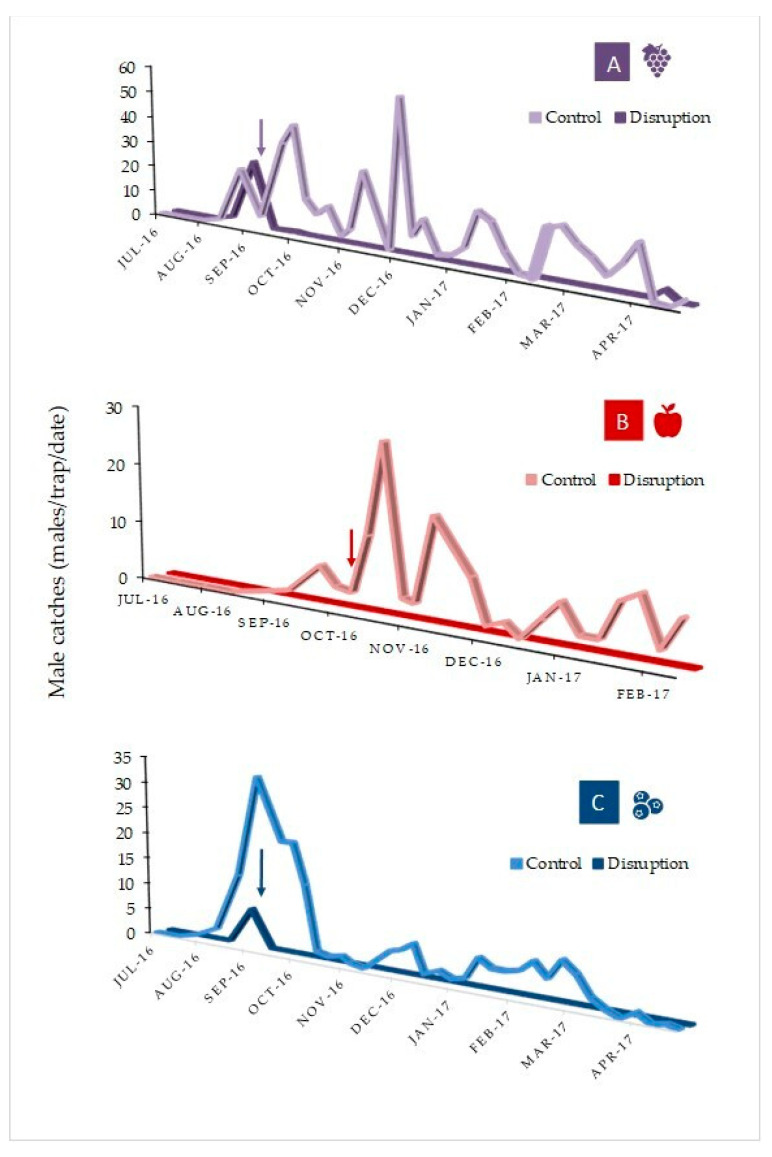
*Proeulia auraria* male captures (avg. males/2 traps*plot/date) in disruption plots (darker color line) and control plots (lighter color line) in a vineyard (**A**: Requinoa), an apple orchard (**B**: San Fernando), and a blueberry orchard (**C**: San Francisco de Mostazal), all located in the O´Higgins region, central Chile, July 2016 through April 2017. The arrow indicates the respective date of SPLAT application.

**Figure 6 insects-12-00625-f006:**
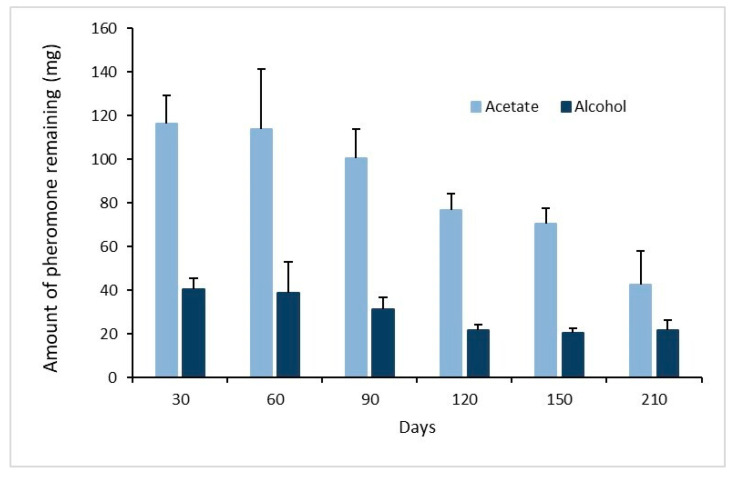
Average amount of pheromone remaining (µg ± SE) per point source (SPLAT matrix loaded as described above, *n* = 5) aged up to 210 days under field conditions.

**Table 1 insects-12-00625-t001:** Mean cumulative *P. auraria* catches in traps (*n* = 3) within aging weeks for the three tested pheromone loads, Nancagua, central Chile, 18–23 February 2015.

	E11-14:OAc + E11-14:OH (μg)		
Aging Weeks	50 + 15	200 + 60	800 + 240
1	13.7 b	24.7 a	26.3 a *
2	26.3 a	20.7 a	20.7 a
3	21.7 a	24.0 a	11.0 b
4	22.7 a	27.7 a	24.0 a
5	8.0 b	27.7 a	23.0 a
6	17.7 b	19.7 b	30.0 a
7	16.0 b	28.3 a	23.7 a
8	13.3 b	21.0 a	22.3 a
9	16.7 b	27.7 a	17.0 b
10	14.7 b	26.0 a	21.0 ab

* Different lowercase letters in a row indicate significant differences (*p* < 0.05) between pheromone doses within an aging time (weeks).

## Data Availability

The data presented in this study are available on request from the corresponding author.

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
