# Peer review of "Development of Monitoring and Mating Disruption against the Chilean Leafroller Proeulia auraria (Lepidoptera: Tortricidae) in Orchards"

_insects, 2021, doi:10.3390/insects12070625_

Round 1

Reviewer 1 Report

The manuscript entitled "Development of Monitoring and Mating Disruption Against the Chilean Leafroller Proeulia auraria (Lepidoptera: Tortricidae) in  
Orchards" is well presented, written and discussed. Overall, I feel the experiments have been well performed, and subsequently, I feel confident about their findings. A minor detail, to me, would be to try to make better figures, at least with the same type of letter that is in the text, I felt the lack of image quality throughout the manuscript. Beisdes that, this is an excellent contribution to the chemical ecology of P. auraria and its ethological control. 

Author Response

We cover both suggestions from reviewer 1. Regarding the type of letter in figures, we changed it, matching now with the type used in the main portion of the article

Figures were improved by using color and bullet points (Figs. 4 and 5).

Reviewer 2 Report

This manuscript “Development of Monitoring and Mating Disruption Against the Chilean Leafroller Proeulia auraria (Lepidoptera: Tortricidae) in Orchards” by Flores et al., provides interesting and useful data from a generally well-planned and well-conducted study. One important shortcoming that should be addressed before publication is the matter in which the statistical analysis is described and reported. It would also be good to incorporate more recent relevant references, concepts, and theories in the discussion. Specific suggestion are provided below, first for statistical issues and then for other concerns and suggestions.

STATISTICAL ISSUES

Section 2.3, line 138: “…the traps were deployed in a completely randomized block design…” Completely randomized design, or randomized complete block design? Given subsequent mention of blocks, the authors appear to mean the latter.

Section 2.3, line 141: “Data were analized by ANOVA… spelling, “analyzed”. Also, it seems that a randomized complete block arrangement was used in the field (previous remark). Was replicate block a factor in the analysis of variance? If so, was it a straight 2-way fixed-effect ANOVA, or was it a mixed model ANOVA with replicate as a random effect?

Section 2.3, line 152: “Data were analyzed by GLM…” The acronym GLM is not previously defined. I presume it means “generalized linear model” (link function with a selected error distribution), but it can also be understood as “general linear model” (ANOVA or regression techniques with a Gaussian distribution and estimated by ordinary least square regression). Saying a that a generalized linear model was was used is insufficient without saying what error distribution was used. It would also be better to say what software and what procedure, package, and/or function was used (for this and all statistical procedures).

Section 2.6, line 191: “Fruit damage was evaluated at harvest by… contrasting the control vs. the MD treatment by using the test for the equality of proportions (p ≤ 0,05) [23].”

Section 3.1, line 204 “The field trial… showed significantly larger P. auraria male catches (p = 0.003)… Need to show the test statistic (for example, F value) and the associated degrees of freedom in addition to the P value.

Line 216: Same issue and remark as above

Lines 228 to 230: Here F and p values are provided for two main effects and an interaction. Degrees of freedom associated with the three effects are still missing. The reported F values do not clarify the ambiguity from line 152 above. The authors note a positive interaction, then apparently test for differences of means of the two main effects. This is questionable in terms of correct statistical practice, and does not directly request the question of applied interest. I suggest testing the lure differences with the full model (as the authors did), then regressing males captured vs. aging time for each lure separately. There will, of course, be no significant slope.

STATISTICAL CONCERNS AND SUGGESTIONS

Table 2: Place replace this with a three-panel scatterplot, similar to Fig. 5.

Line 75: “…the tufted apple bud moth (TBM)… This acronym is defined here, and used for the second and final time on line 335. Please write out the common or scientific name to help those less familiar with your agroecosystem.

Lines 80-82: “In fact, recently this blend was successfully tested in Southern areas of Chile [20]. Additionally, preliminary data obtained from small-scale field trials suggested that P. auraria males might be susceptible to mating disruption [21]” These two references seem to be important precursors to the present paper, but they are not accessible by Internet. Does the following reference provide the same content as reference 21? If so, then is would be preferable to cite that. 

  • Valverde-Rodriguez, A. 2019. FORMULACIÓN SPLAT PARA EL CONTROL DE Proeulia auraria (LEPIDOPTERA: TORTRICIDAE) A TRAVES DEL MÉTODO DE CONFUSIÓN SEXUAL EN FRUTALES Splat formulation for the control of Proeulia auraria (Lepidoptera: T. Revista Investigación Agraria 1: 67-75. https://doi.org/10.47840/ReInA20199

Line 178: Tell the reader why was SPLAT placed in 30 ml plastic cups for the mating disruption trial.

Section 4, Lines 313-314: “It has been discussed in the literature whether or not the whole blend is needed for moth monitoring or mating disruption, or if a partial blend is sufficient [24].” Carde and Minks 1995 is relevant, but too old to be used as a stand-alone reference concerning the whether the most complete blend is the best blend. More recent papers include:

  • Evenden, M. L., G. J. R. Judd, and J. H. Borden. 1999. Pheromone-mediated mating disruption of Choristoneura rosaceana: is the most attractive blend really the most effective? Entomol. Exp. Appl. 90: 37-47.
  • Lapointe, S. L., L. L. Stelinski, T. J. Evens, R. P. Niedz, D. G. Hall, and A. Mafra-Neto. 2009. Sensory imbalance as mechanism of orientation disruption in the leafminer Phyllocnistis citrella: elucidation by multivariate geometric designs and response surface models. J. Chem. Ecol. 35: 896-903.
  • Higbee, B. S., C. S. Burks, and R. T. Cardé. 2017. Mating disruption of the navel orangeworm (Lepidoptera: Pyralidae) using widely spaced aerosol dispensers: is the pheromone blend the most efficacious disruptant? J. Econ. Entomol. 110: 2056-2061.

Lines 328-332: “Regarding the study to determine septa replacement timing, reports for the duration of lures targeting tortricids are very variable (from a few weeks to several months), depending mostly on the species, lures, and pheromone components [27]”. Reference 27, Hofmeyer and Burger 1995 is an interesting paper about an experimental device developed as an alternative to septa lures, but I cannot see where that paper supports this sentence. The subsequent paper, by Biever and Hoffsteter in 1989, is unavailable to me. The results section simply says that there are significant difference in males captured among weeks; it does not address loss of attractiveness of time or lack thereof, and variability. This part of the discussion needs to state more clearly what was found (slope of capture vs. aging time not significantly different from 0 for any of the three treatments, but considerable week-to-week variability in all treatments). If the authors wish to state that there is precedent for such longevity in septa lures in the field, better literature support needs to be provided.  

Lines 387-389: “The literature indicates that the mechanisms in tortricids are either non-com-petitive (e.g. sensory adaptation) or competitive based on dispenser attraction (e.g. false-trail following) [39], but the latter has been identified to be more important in most tortricid species [22].” Reference 39, Miller and Gut 2015, deals with a broader range of species than tortricids but also provides a broader range of scenarios with the Tortricidae. Reference 22, Stelinski et al. 2007, provide a number of examples in which sexual communication in tortricid species was suppressed by a high density of pheromone point sources, but does not explicitly support the generalization that competitive mechanisms are more important in most tortricid species. In fact, the 2007 Stelinski et al paper describing the effect of a SPLAT formulation on the Oriental fruit moth Grapholita molesta, discuss evidence that the mechanism in that study did not closely approach the SPLAT droplets in the first 17 days of the trial, but did so after 17 days. The authors at that time classified both situations as “competitive” because there was evidence that G. molesta initiated orientation along plumes but did not reach the source under that circumstance. However, researchers in the same Michigan State group in a subsequent study compared the impact of monitoring lures and mating disruption dispensers on G. molesta sexual communication and conclude that that mating disruption was competitive for the monitoring lures, but non-competitive for the highly successful mating mating disruption disspensers (Reinke et al. 2014, see below). The last full paragraph in the Miller and Gut 2015 discussion of mechanism further remarks on competitive and non-competitive mechanisms in G. molesta and historical misunderstanding of the data. Recognizing an evolving understanding of mating disruption is important in both developing mating disruption systems and in providing context for such research to a wider scientific readership.

  • Reinke, M. D., P. Y. Siegert, P. S. McGhee, L. J. Gut, and J. R. Miller. 2014. Pheromone release rate determines whether sexual communication of Oriental fruit moth is disrupted competitively vs. non-competitively. Entomol. Exp. Appl. 150: 1-6.

Author Response

We addressed and anwsered all criticism/suggestions from reviewer 2, clearly provided to improve our article. Changes we made include clearing out the statistical analysis and giving a better presentation of results (Table 1), incorporating more recent references for discussión, and replacing two of them for new ones  available/precise regarding the stated concepts. 

Addressing Statistical issues

Regarding section 2.3.,lines 138-152, the reviewer is right; it was a randomized complete block design, so we rephrase those sentences using that denomination. We also explicit it was conducted a straight 2-way fixed-effect ANOVA. GLM stands for ¨General linear model" and we eliminate the acronym in the methodology, replacing it for the complete phrase.

Section 2.6, line 191: we changed the sentence as suggested.

Section 3.1, line 204 and 216, we added the F-value and df in both cases.

Line 228-230. The reviewer is right. Since we found a significant intearction, it is wrong to present the contrast for the factors independently. So, we now only include the statistics for the interaction analysis.

Addressing statistical concerns and Suggestions

Table 2, in fact, Table 1. The criticism is right, so we now present only results for the interaction, i.e. the contrast between captures by pheromone load within a week.

Regarding the mention of the acronym TBM in line 335, it was added the meaning and the respective scientific name

Regarding the comment for Lines 80-82: reference 20 (in Spanish) is actually available through Google Academic (http://cybertesis.uach.cl/tesis/uach/2018/fak.12e/doc/fak.12e.pdf), so we kept this one. We agree with the reviewer to replace the original reference 21 (in fact not available on the WEB; we cited it because we have the printed version) by the one he/she suggested published in Rev. Invest. Agr,

Regarding Line 178. we added a short explanation why we used 30 mL cups to deploy the SPLAT in the field

Regarding the criticism stated about the 313-314 statement, we agree with the reviewer and added the three more recent references (she/he provided) explaining that in some studies a biene blend (as our case) has been successfully used/tested for mating disruption in several lepidopteran pests.

Regarding the criticism stated about lines 328-332, we did three changes, as follows:

a.- Paper by Biever and Hosteter (1989). We neither found the original paper, but the abstract (available on the WEB) states that 10 weeks aged septa works as well as fresh material, coinciding with our findings, that´s why we cited and we think it should be kept as reference.

b.- We replaced the paper by Hofmeyer and Burger (1995) (it actually did not support our original statement as the reviewer said) by the one published by Knight (2002) that explicitily states the importance of lure type and pheromone load (the focus of our discussion in this paragraph) on lure longevity for tortricid monitoring.

c.- We add to the paragraph the sentence the reviewer include in his/her review stating our main findings regarding septa longevity, i.e. the 200 + 60 ug blend was the selected one.

Regarding last criticism (lines 387-389), we agreed with that and now included the findings by Reinke et al (2014), mentioning that either competitive and non-competitive disruption mechanisms can operate on male moths depending on pheromone atmospheric concentration

Reviewer 3 Report

This is a very important study designed to develop monitoring and mating disruption technique for Chilean leafroller Proeulia auraria. The paper is very well written, it provides great background information, the methods are described in great detail and are easy to follow. Results and discussion are also well written. I have a few very minor comments:

Lines 13 and 23: I would change "increasing pest" to "growing pest"

Line 14: remove word "compound" 

Line 34: what is male disruption? Did you mean "mating disruption"?

Line 98: change to "programs using Bt..."

Line 180: I am surprised that you used blank SPLAT for control instead of leaving plots completely untreated. Maybe explain the reason in the methods as this is unconventional. 

Author Response

We cover all suggestions from reviewer 1.

As suggested, in Lines 13 and 23 we replaced the word "increasing" by "growing" as suggested by reviewer 1

In Line 14 we did not remove the word "compound" because we want to emphasize we were working with several pheromone components (not just one) to prepare and test different blends

In Line 34: we changed male disruption by "mating disruption" to clarify the concept, as suggested

Regarding the line 180´s suggestion for explaining the use of blank SPLAT in our field trials, we added two recent references where the researchers also applied blank splat for control treatment.